# Safety and Tolerability of COVID-19 Vaccines in Patients with Cancer: A Single Center Retrospective Analysis

**DOI:** 10.3390/vaccines10060892

**Published:** 2022-06-02

**Authors:** Amedeo Nuzzo, Simona Manacorda, Enrico Sammarco, Andrea Sbrana, Serena Bazzurri, Federico Paolieri, Fiorella Manfredi, Chiara Mercinelli, Marco Ferrari, Giulia Massaro, Adele Bonato, Alessia Salfi, Luca Galli, Riccardo Morganti, Andrea Antonuzzo, Chiara Cremolini, Gianluca Masi

**Affiliations:** 1Unit of Medical Oncology 2, Azienda Ospedaliero-Universitaria Pisana, Santa Chiara Hospital, 56125 Pisa, Italy; manacorda.simona@gmail.com (S.M.); enricosammarco1992@gmail.com (E.S.); serena.baz@gmail.com (S.B.); federico.paolieri@gmail.com (F.P.); manfredifiorella@gmail.com (F.M.); chiara.mercinelli@gmail.com (C.M.); marco.ferrari0201@gmail.com (M.F.); giulia21massaro@gmail.com (G.M.); adelebonato@gmail.com (A.B.); alessiasalfi@gmail.com (A.S.); lugal71@yahoo.it (L.G.); chiaracremolini@gmail.com (C.C.); gianlucamasi72@gmail.com (G.M.); 2Department of Surgical, Medical and Molecular Pathology and Critical Area Medicine, Azienda Ospedaliero-Universitaria Pisana, 56125 Pisa, Italy; andreasbrana89@gmail.com; 3Section of Statistics, University Hospital of Pisa, 56125 Pisa, Italy; r.morganti@ao-pisa.toscana.it; 4Unit of Medical Oncology 1, Azienda Ospedaliero-Universitaria Pisana, Santa Chiara Hospital, 56125 Pisa, Italy; automezzo69@gmail.com

**Keywords:** COVID-19, COVID-19 vaccines, cancer patients, COVID-19 vaccine safety

## Abstract

Background: Severe acute respiratory syndrome coronavirus 2 disease (COVID-19) has caused a worldwide challenging and threatening pandemic. Multinational, placebo-controlled, observer-blinded trials were conducted since the beginning of pandemic because safe and effective vaccines were needed urgently. In most trials of COVID-19 vaccines patients affected by malignancies or on treatment with immunosuppressive drugs were excluded. Patients and methods: A retrospective monocentric study was conducted at Medical Oncological Unit of Santa Chiara Hospital (Pisa, Italy) in this subset of population to investigate safety and tolerability of COVID-19 vaccines; 377 patients with solid tumor on treatment were enrolled. Vaccine-related adverse events were recorded using a face-to-face questionnaire including a toxicity grading scale. Most of the patients (94%) received mRNA vaccine as indicated by Italian health ministry guidelines. Mean age was 66 years (range 27–87), 62% of the patients were older than 65 years and 68% had at least one additional comorbidity. The majority (86%) of patients were in a metastatic setting and 29% received immunotherapy-based treatment. For statistical analysis, multivariate binary logistic regression models were performed and linear regression models were applied. Results: Adverse events were mild and transient and ended in a few days without any sequelae. No severe or uncommon adverse events were recorded. In multivariate analysis, we found that the female sex was associated with a greater risk of more severe and longer lasting adverse events, and a higher risk of adverse events was found for patients treated with immunotherapy. Conclusions: Our results demonstrate that COVID-19 vaccines were safe and well-tolerated in this population of patients being treated for solid tumors.

## 1. Introduction

Since December 2019, the severe acute respiratory syndrome coronavirus 2 (SARS-CoV-2) infection has affected more than 500 million people and caused more than 6 million deaths worldwide [1], prompting the World Health Organization to declare a pandemic on March 2020. In Europe, there have been more than 200 million cases and more than 2 million deaths, while in Italy there were about 17 million people affected and 166 thousand deaths since the beginning of pandemic [1].

From then on, a substantial and growing number of different COVID-19 vaccines was investigated in clinical trials, and most of them showed that vaccines decrease incidence and complications of COVID-19 in healthy adults, carrying to their authorization and approval in several countries worldwide; these include nucleic acid-based, viral-vector, inactivated or recombinant protein vaccines [2]. These have been shown to be safe in the general population; the most common adverse events described were pain, swelling, redness in the injection site, and a flulike syndrome with fever, chills, headache.

Patients with active malignancies may be at higher risk of mortality, prolonged and severe infection and complications from SARS-CoV-2 disease than non-cancer patients, due to their age, disease, cancer-related treatment and medical comorbiditie [3,4]. However, most trials of COVID-19 vaccines excluded patients with prior history of cancer, treated with immunosuppressive therapies, with immunodeficiency and lack of stable disease; consequently, data on the safety, tolerability and efficacy of the vaccines for these patients are currently limited.

Most national and international guidelines recommend COVID-19 vaccination for patients with cancer [5,6,7], highlighting the need to prioritize those with cancer on active treatment, planned to start therapy and in a period of less than six months post therapy completion. Desai et al. [8] emphasize the prioritization for vaccination of cancer patients on active treatment at the earliest available opportunity, regardless of anticancer therapy timing, even in patients enrolled in clinical trials and for their caregivers.

There are no additional safety issues suggesting the adverse events increase in patients with cancer, although rare case reports have been reported [9]; however, strong safety data of COVID-19 vaccine in cancer population are currently missing.

Indeed, COVID-19 vaccines have been shown to be well-tolerated in phase 3 trials of healthy subjects and in few series of cancer patients. We conducted this study with the aim to investigate safety and tolerability of SARS-CoV-2 vaccines and to confirm data already presented in the literature in a cohort of oncological patients on treatment at our cancer centre.

## 2. Patients and Methods

### 2.1. Study Designand Participants

We did an observational, longitudinal, retrospective study conducted at Medical Oncological Unit of Santa Chiara Hospital in Pisa, Italy. Patients were enrolled between 10 May and 11 June 2021.

We included patients older than 18 years of age, with solid malignancies undergoing active anticancer treatment, who had completed the full COVID-19 vaccination course at least 7 days after the last jab. We included patients who had received COVID-19 vaccines available in Italy at that time: Pfizer Comirnaty, Moderna Spikevax, AstraZeneca Vaxzevria and Janssen Ad26.COV2-S. Patients receiving only hormonal adjuvant treatment were excluded.

### 2.2. Data Collection and Study Instrument

From medical records we collected the following baseline patients’ characteristics: age, sex, ECOG PS (Eastern Cooperative Oncology Group Performance Status), comorbidities, type of malignancy, TNM stage, treatment setting, type of ongoing anticancer medication, vaccine administered, previous COVID-19 infection.

A detailed questionnaire was administered face-to-face by doctor to the patients in our clinic. This survey evaluated a toxicity grading scale regarding any adverse events occurring after each vaccine dose. We considered as vaccine-related adverse events those occurring within 7 days after vaccination, according to the literature data [10].

We used a standardized grading system developed by the Food and Drug Administration for people enrolled in vaccine clinical trials [11]; parameters included were local (pain, redness and swelling around injection site, lymphadenopathy) and systemic (asthenia, headache, fever, muscle or joint pain, vomit, diarrhoea, anaphylaxis and chills).

These parameters were graded from 0 to 4 according to the standardized grading system as follow: pain at the injection site (mild: does not interfere with activity; moderate: repeated use of non-narcotic pain reliever >24 h or interferes with activity; severe: any use of narcotic pain reliever or prevents daily activity), redness (mild: 2.0 to 5.0 cm in diameter; moderate: >5.0 to 10.0 cm in diameter; severe: >10.0 cm in diameter), swelling (mild: does not interfere with activity; moderate: interferes with activity; severe: prevents daily activity), lymphadenopathy, asthenia, chills and muscle or joint pain (mild: no interference with activity; moderate: some interference with activity not requiring medical intervention; severe: prevents daily activity and requires medical intervention), febricula (mild: body temperature greater than 37 but lower than 38 degrees), fever (mild: 38.0–38.4; moderate: 38.5–38.9; severe: 39.0–40; grade 4: >40), vomit (mild: 1 to 2 times in 24 h; moderate: >2 times in 24 h; severe: requires intravenous hydration), diarrhoea (mild: 2 to 3 loose stools in 24 h; moderate: 4 to 5 loose stools in 24 h; or severe: 6 or more loose stools in 24 h); grade 4 for all events indicated an emergency department visit or hospitalization.

### 2.3. Endpoint and Statistical Analysis

The endpoint of this study was to report the safety profile of COVID-19 vaccines in this population.

Categorical data were described by absolute and relative frequency, continuous data by mean and standard deviation.

Comparisons between categorical factors were carried out by Chi square test and z-test for two proportions.

To analyze demographic and clinical factors as age, sex, ECOG PS, comorbidity, setting, treatment with immunotherapy, vaccine type, influencing dichotomous outcomes as local adverse events (no, yes) and systemic adverse events (no, yes) multivariate binary logistic regression models were performed, while to evaluate the impact of the same factors on the continuous outcomes as local/systemic adverse events grade and local/systemic adverse events duration, multiple linear regression models were applied.

Results of the multivariate models were expressed by regression coefficient, odds ratio associated to 95% confidence interval (CI) for the logistic models and partial correlation coefficient for the linear models.

Significance was fixed at 0.05 and all analysis were carried out using Statistical Package for Social Science (SPSS) technology v.27 (IBM Corporation, New York, NY, USA).

### 2.4. Ethics

This study was performed in line with the principles of the Declaration of Helsinki. Approval was granted by local Ethics Committee. Informed consent was obtained from all individual participants included in the study.

## 3. Results

### 3.1. Demographic and Clinical Features of Participants

A total of 377 patients was enrolled. Patients’ characteristics are summarized in Table 1. Mean duration of follow-up after the last dose of vaccine was 22 days.

The mean age was 66 years and 62.3% (235/377) of participants were older than 65 years. Most patients (91.3%) had an ECOG PS of 0 or 1; half of the participants were female.

Of the patients, 68.4% had at least one coexisting condition in addition to cancer, the most frequent medical comorbidity was cardiovascular disease (reported in 48% of patients), followed by metabolic/endocrine disorders (except diabetes; 22.5%) and diabetes (11.4%).

The most frequent oncological malignancies were thoracic (27.3%) and gastrointestinal (27.3%), followed by breast and gynaecological (20.7% together).

At the time of vaccination, 52 patients (13.7%) were receiving a neoadjuvant or adjuvant anticancer treatment for localized disease, while 213 (56.5%) were receiving first line treatment for metastatic disease, and 112 (29.8%) for second or further therapy lines.

Among these 377 patients, 33 (8.8%) had a previous SARS CoV-2 infection: only two of them (6%) developed severe disease, 6 (18.2%) moderate illness, 13 (39.4%) mild disease and 12 (36.4%) an asymptomatic form, according to NIH classification [12].

Most patients received a COVID-19 mRNA (messenger RNA) vaccine (48.5% of participants received the Moderna Spikevax and 47.8% Pfizer Comirnaty vaccine), and just 3.7% received viral vector vaccines (Astra-Zeneca Vaxzevria and Janssen Ad26.COV2-S).

### 3.2. Safety

#### 3.2.1. First Dose

After the first dose of vaccine 245/377 (67.4%) patients developed at least one adverse event. The most common adverse event was injection site local pain, reported by 211 patients (56%; only two patients developed grade 3 pain); mean pain duration was 2 days. Among local adverse events: swelling in injection site 44 patients (11.7%), redness 26 (6.9%) and locoregional lymphadenopathies 4 (1.1%); regarding these above-mentioned adverse events, no grade 3 or 4 were reported.

Within systemic adverse events, we reported: asthenia 74 patients (19.7%; only three patients developed grade 3 asthenia), febricula (body temperature greater than 37 but lower than 38 degrees) 8 (2.1%), fever (temperature greater than 38 degrees) 20 (5.3%), chills 15 (4%) and muscle or joint pain 23 (6.1%). One patient had an immediate cutaneous allergic reaction after vaccine administration (resolved spontaneously without any life-threatening consequences); gastrointestinal adverse events (vomiting and diarrhoea) were noted in less than 3% of patients.

The majority of adverse events had resolved spontaneously, only 42/377 (11.1%) patients used symptomatic treatment (e.g., pain relievers, antipyretics, etc.).

#### 3.2.2. Second Dose

Among 354 patients received the second vaccine dose, 257 (72.6%) developed at least one adverse event; 23 patients did not receive the second dose of vaccine (6 patients underwent to Janssen vaccine, 17 patients to a single dose of vaccine due to a previous COVID-19 within 6 months).

The most common adverse event was injection site local pain, reported by 184 patients (52%; only 1 patient developed grade 3); mean pain duration was 2 days.

Among local adverse events, swelling in injection site was noted in 51 patients (14.4%), redness in 31 (8.7%) and locoregional lymphadenopathies in 4 (1.1%); no grade 3 or 4 adverse events were reported. Between systemic adverse events, asthenia was reported in 112 patients (31.6%; 7 patients developed grade 3 asthenia), febricula in 30 (8.5%), fever in 76 (21.5%), chills in 64 (18.1%) and muscle or joint pain in 74 (20.9%). After second vaccine dose three patients had immediate hypersensitivity cutaneous reactions (resolved spontaneously without any life-threatening consequences); vomiting was reported in 9 (2.5%) and diarrhoea in 10 (2.8%).

Symptomatic treatment (e.g., pain relievers, antipyretics, etc.) was required in 101/354 patients (28.5%).

#### 3.2.3. Comparison between Percentages of Adverse Events in Our Cancer Population and in a Healthy Individuals’ Population

As shown in Figure 1, we compared the percentages of each adverse event after each vaccine dose, between the cancer patients population described in our study and a large sample of healthy individuals included in the survey of Rosenblum et al. [13]; statistically significant differences emerged in most of them, as reported in Table 2.

#### 3.2.4. Risk Factors for Adverse Events

In the multivariate logistic models, an age younger than 65 years [odds ratio (OR), 0.55; 95% CI, 0.32–0.94; *p* = 0.029] and female sex (OR, 1.84; 95% CI, 1.17–2.89; *p* = 0.008) were both considered significant risk factors for local adverse events after the first dose, but they did not reach statistical significance with regard to systemic adverse events after the first dose.

mRNA vaccines were associated with higher risk of local adverse events after the second dose (OR, 0.19; 95% CI, 0.03–0.97; *p* = 0.046), although we only had 8/354 individuals with non-mRNA vaccines who underwent second vaccination dose, so our power was rather low for this particular conclusion. Female sex (OR, 2.31; 95% CI, 1.45–3.69; *p* < 0.001), ECOG PS 0 (OR, 0.68; 95% CI, 0.46–1; *p* = 0.050) and treatment with immune checkpoint inhibitors (ICIs) (OR, 1.67; 95% CI, 1.01–2.76; *p* = 0.047) were associated with higher risk of systemic adverse events after the second dose too (Table 3).

Considering multiple linear regression based on the adverse events’ grade, female sex was significantly associated with a greater risk of more severe local adverse events after both the first (*p* = 0.001) and the second dose (*p* = 0.003) and with a greater risk of more severe systemic adverse events after the first (*p* = 0.021) and the second dose (*p* = 0.004). Furthermore, use of mRNA vaccine was associated with greater risk of more severe local (*p* = 0.025) and systemic (*p* = 0.048) adverse events after the second dose; non-metastatic setting was considered a risk factor for systemic adverse events after the second dose (*p* = 0.046) (Table 4).

Moreover, in multiple linear regression based on the adverse events’ duration, female sex was confirmed as a significant risk factor for longer lasting local (*p* = 0.022) adverse events after the first dose and for both local (*p* = 0.002) and systemic (*p* = 0.003) adverse events after the second dose. Use of viral vector vaccines was associated with greater risk of longer local adverse events after the first dose (*p* = 0.035), while an ECOG PS greater or equal to 2 with higher risk of longer systemic adverse events after the first dose (*p* = 0.008).

Non-metastatic setting of disease proved to be a risk factor for longer local adverse events after the second dose (*p* = 0.017); use of ICIs was significantly associated with longer systemic adverse events after the second dose (*p* = 0.016) (Table 5).

A previous SARS CoV-2 infection was significantly associated with a greater risk of systemic adverse events (*p* = 0.034) and of higher-grade systemic adverse events (*p* = 0.006).

#### 3.2.5. Effects of Previous COVID-19 on Adverse Events

A further analysis focusing on patients with prior SARS-CoV-2 infection was performed. Among 33 patients with prior COVID-19, 15 developed local adverse events and 14 systemic adverse events.

In the univariate analysis comparing patients with and without prior COVID-19, a statistically significant association was found between prior COVID-19 and the development of systemic adverse events after first vaccination dose (*p* = 0.034); moreover, prior COVID-19 was associated with a medium higher grade of non-serious adverse events (*p* = 0.006) (Table 6).

## 4. Discussion

Patients enrolled in our study represent a completely different population compared to those included in vaccine registered trials: in Moderna [14] and Pfizer [15] trials just 21% and 20.5% patients, respectively, had at least one comorbidity, median age was respectively 52 and 51.5 years, patients older than 65 years were respectively 25.3% and 22%. Recently a post-hoc subgroup analysis of global phase 3 Comirnaty vaccine study regarding patients with a history of malignancy was performed, but this study included also patients with benign tumors and patients receiving systemic immunosuppressive treatment were excluded [16]. Instead, in our study every patient had a diagnosis of active cancer on treatment, 68.4% of patients had at least one further coexisting condition and 62.3% was older than 65 years. In this subset of patients, at higher risk of severe complications of COVID-19 that may lead to death, vaccination is the only weapon capable to prevent it.

Our findings show that COVID-19 vaccines are safe in cancer patients, and incidence of local and systemic adverse events seems lower than we expected if compared to vaccine trials in healthy population. This should be related to the older median age of population in our trial: we know from mRNA vaccine trials that adverse events are distributed according to age, being more common in younger (<65 years old) than older [14,15]. Furthermore, this difference in reactogenicity, meant as the capacity of the vaccine to induce adverse reactions, could be explained by a higher resilience and immune response decline in older people, and particularly in patients with cancer due to concomitant overlapping symptoms and therapies. Another possible explanation is that cancer patients have a reduced immunogenicity to vaccines compared to healthy people, as shown in recent published studies [17,18], and this could lead to a lower rate incidence of vaccine-related adverse events.

Consistently with vaccine trials, incidence of adverse events is higher after the second dose than after the first one [13]. The majority of adverse events were mild and transient, resolving in a few days without sequelae. The most common adverse event was injection site local pain, which usually occurs after injection by a needle. No serious or uncommon adverse eventswere found, just four cutaneous hypersensitivity reactions occurred, all resolved spontaneously without any life-threatening consequences. Although we did not include an unvaccinated control arm, it is likely that most of the adverse events observed were caused by vaccination. Thus it is fair to consider most or all of our observed adverse events as side effects of vaccination.

A recent study by a British group [19] including oncological patients younger than 65 years old confirmed the favorable safety profile after the first dose of COVID-19 vaccines. Similar results were reported by a group in Israel, without severe adverse events, either life threatening or requiring hospitalization, after the full vaccination course [18]. Funakoshi et al. [20] confirmed the acceptable toxicity profile of the Comirnaty vaccine in a small sample of Japanese population with cancer; their data also indicate that serum antibody titer against Spike protein-S1 were significantly lower in patients treated with chemotherapy and ICIs compared to healthy volunteers, underlining the state of chronic immunosuppression of cancer patients due to malignancy itself regardless of the treatment administered.

The survey of Shulman et al. [21] reported no significant differences between patients with and without cancer in the frequency of adverse events after both vaccination jabs; symptoms were more likely reported by women and younger patients, according to our data; however, in this study only few patients with cancer were receiving active treatment.

Trillo Aliaga et al. [22] pointed out to the favorable safety profile of COVID-19 vaccines even in patients treated in early-phase clinical trials with novel antineoplastic agents; moreover, the authors did not find any association between the incidence of adverse events and the timing between the vaccination and the previous cancer therapy.

In our study, the multivariate analysis showed a statistically significant association between female sex and onset of local and systemic adverse events after each vaccination dose: those adverse events were higher in grade and longer in duration in females. This analysis was not included in vaccine trials, but a vaccine safety analysis recently conducted in a real-life study in the United States [23] and the European EUDRAvigilance database [24] showed that the majority of adverse events reports concerned women, according with the above-mentioned studies of So et al. [19] and Shulman et al. [21].

A possible hypothesis is that women, for biological and hormonal reasons, tend to have a stronger humoral immunity and a better immune response to vaccines than men, and a stronger immune response to vaccine could be associated to a higher rate of adverse events. As a matter of fact, a recent study showed that the presence of female sexual hormone estradiol led to a greater and better vaccine-induced antibody response to influenza vaccine [25].

Furthermore, we found that patients on treatment with ICIs are associated with a higher rate of onset and duration of systemic adverse events after the second dose compared to people on treatment with other anticancer treatments, with a trend in a higher grading of these adverse events. Vaccination could act as an additional trigger to immune response in patients treated with ICIs, leading to a greater rate, duration and grade of adverse events. However, we did not find an increasing frequency of immune-related adverse events. We performed other categorizations of the variable “anticancer treatment”, but no statistical significance emerged.

Moreover, we found that mRNA vaccines are associated with a higher grade of both local and systemic adverse events after the second dose. This, though with the limit of the small number of patients undergone adenovirus vaccine, is consistent with published data: adenovirus vaccines are more likely to give more frequent adverse events and higher in grade after the first dose [26], instead mRNA vaccines after the second.

A weak association was found in terms of onset of systemic adverse events after the second vaccination dose and an ECOG PS equal or more than 2: a potentially explanation could lie in the fragility of this setting of patients, which could be more likely to report adverse events of lower intensity.

Focusing on people with prior SARS-CoV-2 infection, despite the small sample size of this part of population, an association was found between prior COVID-19 and development of non-serious systemic adverse events, which were higher grade compared with people without previous infection. Indeed, considering the immunity induced by natural infection, patients with prior SARS-CoV-2 infection might be at higher risk of adverse events after first vaccine jab due to immune memory and inflammatory reactions, consistently with a similar study performed in healthy individuals [27]. However, in our study no serious adverse events were reported in this sample.

Our study has some limitations: first, the retrospective and mono-institutional nature of this analysis and the number of patients enrolled. Second, a substantial part of our patients is on treatment with analgesic drugs, corticosteroids, etc that could hide any adverse events (it was not possible to determine the impact of these medications on adverse events development). Last, we do not have any information about the efficacy of vaccines in our study population: no nasopharyngeal swabs or SARS-CoV-2 antibody serologic tests were performed after vaccination; however, it should be emphasized that the main objective of the current analysis was to assess the safety of vaccine in cancer population.

So far, our study suggests that SARS-CoV-2 vaccination is safe in cancer patients under active oncological treatment, and it should be prioritized and encouraged in this specific subgroup of patients. However, further prospective studies are needed to confirm our results and to better determine efficacy, immune response and duration of protective immunity to COVID-19 in patients with cancer, and also to investigate the correlation between humoral and cell-mediated response to COVID-19 vaccines and protection against SARS-CoV-2 infection.

## Figures and Tables

**Figure 1 vaccines-10-00892-f001:**
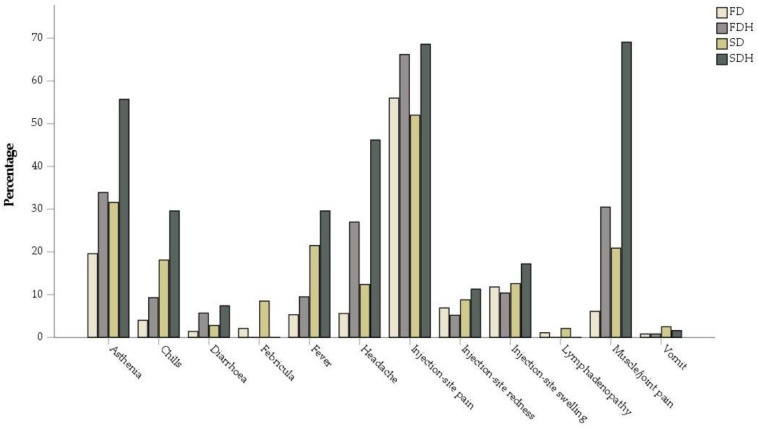
Comparison between percentages of adverse events in our study’s population and Rosenblum et al.’s population on healthy individuals [13]. FD = first dose (*n* = 377); SD = second dose (*n* = 354); FDH = first dose on healthy individuals (*n* = 5,674,420), SDH = second dose on healthy individuals (*n*= 6,775,515).

**Table 1 vaccines-10-00892-t001:** Patients’ demographic and clinical characteristics.

Characteristic	Total (*n* = 377)
Median Age (range)	66 (27–87)
Patients older than 65 years	235 (62.3)
Sex—no. of patients (%)	
Male	191 (50.7)
Female	186 (49.3)
ECOG PS—no. of patients (%)	
0	211 (56)
1	133 (35.3)
2	28 (7.4)
3	5 (1.3)
Non oncological comorbidities—no. of patients (%)	
At least one condition	258 (68.4)
Cardiovascular disease (includes hypertension)	181 (48)
Lung disease	28 (7.4)
Eye, ear, nose and throat disease	14 (3.7)
Gastroenteric and epatic disease	37 (9.8)
Genito-urinary disease	37 (9.8)
Musculoskeletal and cutaneous disease	21 (5.5)
Neurologic disease	15 (3.9)
Reumatological disease	13 (3.4)
Endocrinological Disease	85 (22.5)
Diabetes	43 (11.4)
Psychiatric disease (includes dementia)	19 (5)
Obesity (Body Mass Index > 30 kg/m^2^)	36 (9.5)
Type of malignancy—no. of patients (%)	
Women’s cancer (breast and gynecological)	78 (20.7)
Urological cancer (renal, prostate, testicular, bladder)	61 (16.2)
Skin cancers	21 (5.6)
Thoracic cancers	103 (27.3)
Gastrointestinal cancers	103 (27.3)
Head and neck cancer	5 (1.3)
Others	6 (1.6)
TNM staging—no. of patients (%)	
III	52 (13.79)
IV	325 (86.21)
Treatment setting—no. of patients (%)	
Neoadjuvant	17 (4.5)
Adjuvant	35 (9.2)
First line	213 (56.5)
Second line or following	112 (29.8)
Anticancer treatment—no. of patients (%)	
Chemotherapy	127 (33.7)
Target therapy	104 (27.6)
Immunotherapy (includes immunotherapy combinations)	111 (29.4)
Chemotherapy plus target-therapy	35 (9.3)
Previous COVID-19—no. of patients (%)	
no	344 (91.2)
yes	33 (8.8)
Vaccine administered—no. of patients (%)	
Astrazeneca	8 (2.1)
Janssen	6 (1.6)
Moderna	183 (48.5)
Pfizer	180 (47.8)

**Table 2 vaccines-10-00892-t002:** Comparison between percentages of adverse events in our study’s population and Rosenblum et al.‘s population on healthy individuals [13]. FD = first dose (*n* = 377); SD= second dose (*n* = 354); FDH = first dose on healthy individuals (*n* = 5,674,420), SDH= second dose on healthy individuals (*n* = 6,775,515). *: *p*-values reaching statistical significancy.

	FD	FDH	SD	SDH	*p*-Values	
					FD vs. FDH	SD vs. SDH
Injection-site pain	56	66.2	52	68.6	<0.001 *	<0.001 *
Injection-site redness	6.9	5.2	8.8	11.3	0.174	0.163
Injection-site swelling	11.8	10.4	12.6	17.2	0.424	0.027 *
Lymphadenopathy	1.1	not available	2.1	not available	/	/
Asthenia	19.6	33.9	31.6	55.7	<0.001 *	<0.001 *
Headache	5.6	27	12.4	46.2	<0.001 *	<0.001 *
Febricula	2.1	not available	8.5	not available	/	/
Fever	5.3	9.5	21.5	29.6	0.007 *	0.001 *
Chills	4	9.3	18.1	29.6	<0.001 *	<0.001 *
Muscle/joint pain	6.1	30.5	20.9	69.1	<0.001 *	<0.001 *
Vomit	0.8	0.8	2.5	1.6	0.999	0.260
Diarrhoea	1.4	5.7	2.8	7.4	<0.001 *	<0.001 *

**Table 3 vaccines-10-00892-t003:** Multivariate binary logistic regression of the clinical factors influencing the categorical outcomes (0: no, 1: yes). Age: (0) <65, (1) ≥65; Sex: (0) M, (1) F; ECOG PS: (0) 0, (1) 1, (2) ≥2; Comorbidity: (0) no, (1) ≤2, (2) ≥3; Setting: (0) adjuvant and neoadjuvant, (1) metastatic; ICIs treatment: (0) no, (1) yes; Vaccine: (0) with mRNA, (1) with viral vector. *: *p*-values reaching statistical significancy.

Outcome	Factor	OR (95% CI)	*p*-Value
Local adverse events after the first dose	Age	0.55 (0.32–0.94)	0.029 *
	Sex	1.84 (1.17–2.89)	0.008 *
	ECOG PS	0.91 (0.64–1.32)	0.634
	Comorbidity	0.85 (0.69–1.06)	0.152
	Setting	0.75 (0.37–1.51)	0.422
	ICIs treatment	0.77 (0.48–1.24)	0.283
	Vaccine	0.39 (0.12–1.25)	0.112
Systemic adverse events after the first dose	Age	0.57 (0.32–1.01)	0.055
	Sex	1.61 (0.98–2.64)	0.060
	ECOG PS	1.13 (0.75–1.72)	0.552
	Comorbidity	0.93 (0.73–1.19)	0.566
	Setting	0.87 (0.45–1.68)	0.674
	ICIs treatment	0.89 (0.52–1.52)	0.671
	Vaccine	1.69 (0.55–5.24)	0.362
Local adverse events after the second dose	Age	0.68 (0.34–1.16)	0.158
	Sex	1.46 (0.92–2.31)	0.105
	ECOG PS	0.81 (0.56–1.18)	0.281
	Comorbidity	1.07 (0.85–1.33)	0.580
	Setting	0.79 (0.39–1.54)	0.471
	ICIs treatment	0.95 (0.58–1.55)	0.848
	Vaccine	0.19 (0.03–0.97)	0.046 *
Systemic adverse events after the second dose	Age	0.95 (0.55–1.63)	0.854
	Sex	2.31 (1.45–3.69)	<0.001 *
	ECOG PS	0.68 (0.46–1)	0.050 *
	Comorbidity	0.87 (0.69–1.10)	0.249
	Setting	0.84 (0.43–1.65)	0.613
	ICIs treatment	1.67 (1.01–2.76)	0.047 *
	Vaccine	0.22 (0.04–1.21)	0.082

**Table 4 vaccines-10-00892-t004:** Multiple linear regression of the clinical factors influencing the continuous outcomes based on the adverse events grade. Age: (0) <65, (1) ≥65; Sex: (0) M, (1) F; ECOG PS: (0) 0, (1) 1, (2) ≥2; Comorbidity: (0) no, (1) ≤2, (2) ≥3; Setting: (0) adjuvant and neoadjuvant, (1) metastatic; ICIs treatment: (0) no, (1) yes; Vaccine: (0) with mRNA, (1) with viral vector. PCC: partial correlation coefficient; *: *p*-values reaching statistical significancy.

Outcome	Factor	PCC	*p*-Value
Grade of the local adverse events after the first dose	Age	−008	0.179
	Sex	0.18	0.001 *
	ECOG PS	−0.03	0.601
	Comorbidity	−0.02	0.686
	Setting	−0.09	0.101
	ICIs treatment	−0.01	0909
	Vaccine	−0.01	0.908
Grade of the systemic adverse events after the first dose	Age	−0.11	0.066
	Sex	0.12	0.021 *
	ECOG PS	0.05	0.360
	Comorbidity	0.08	0.163
	Setting	−0.02	0.756
	ICIs treatment	−0.03	0.534
	Vaccine	0.05	0.343
Grade of the local adverse events after the second dose	Age	−0.10	0105
	Sex	0.16	0.003 *
	ECOG PS	−0.04	0.523
	Comorbidity	−0.01	0.860
	Setting	−0.09	0.086
	ICIs treatment	−0.01	0.893
	Vaccine	−0.12	0.025 *
Grade of the systemic adverse events after the second dose	Age	−0.05	0.391
	Sex	0.16	0.004 *
	ECOG PS	−0.07	0.253
	Comorbidity	0.02	0.685
	Setting	−0.11	0.046 *
	ICIs treatment	0.10	0.055
	Vaccine	−0.10	0.048 *

**Table 5 vaccines-10-00892-t005:** Multiple linear regression of the clinical factors influencing the continuous outcomes based on the adverse events duration. Age: (0) <65, (1) ≥65; Sex: (0) M, (1) F; ECOG PS: (0) 0, (1) 1, (2) ≥2; Comorbidity: (0) no, (1) ≤2, (2) ≥3; Setting: (0) adjuvant and neoadjuvant, (1) metastatic; ICIs treatment: (0) no, (1) yes; Vaccine: (0) with mRNA, (1) with viral vector. PCC: partial correlation coefficient; *: *p*-values reaching statistical significancy.

Outcome	Factor	PCC	*p*-Value
Duration of the local adverse events after the first dose	Age	−0.06	0.315
	Sex	0.12	0.022 *
	ECOG PS	−0.01	0.849
	Comorbidity	−0.03	0.565
	Setting	0.03	0.557
	ICIs treatment	0.05	0.362
	Vaccine	0.11	0.035 *
Duration of the systemic adverse events after the first dose	Age	−0.07	0.240
	Sex	0.02	0.689
	ECOG PS	0.15	0.008 *
	Comorbidity	0.07	0.182
	Setting	0.01	0.884
	ICIs treatment	−0.02	0.765
	Vaccine	0.05	0.330
Duration of the local adverse events after the second dose	Age	−0.05	0.450
	Sex	0.17	0.002 *
	ECOG PS	0.04	0.490
	Comorbidity	−0.04	0.495
	Setting	−0.13	0.017 *
	ICIs treatment	0.08	0.121
	Vaccine	−0.09	0.079
Duration of the systemic adverse events after the second dose	Age	−0.01	0.831
	Sex	0.17	0.003 *
	ECOG PS	0.03	0.576
	Comorbidity	0.05	0.330
	Setting	−0.06	0.289
	ICIs treatment	0.13	0.016 *
	Vaccine	−0.07	0.177

**Table 6 vaccines-10-00892-t006:** Comparison between previous COVID-19 and adverse events after the first dose. Statistics: absolute frequency or mean (sd). *: *p*-values reaching statistical significancy.

	Prior COVID-19		*p*-Value
Adverse event developed	no	yes	
Local adverse events			0.108
no	138	18	
yes	206	15	
Systemic adverse events			0.034 *
no	257	19	
yes	87	14	
Medium grade of local adverse events	0.22 (0.25)	0.19 (0.26)	0.448
Medium grade of systemic adverse events	0.06 (0.14)	0.13 (0.18)	0.006 *
Medium duration of local adverse events	0.49 (1.02)	0.50 (1.07)	0.947
Medium duration of systemic adverse events	0.17 (0.64)	0.38 (0.60)	0.062

## Data Availability

The data supporting the findings of this study are contained within the article.

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
