# Peer review of "Safety and Tolerability of COVID-19 Vaccines in Patients with Cancer: A Single Center Retrospective Analysis"

_vaccines, 2022, doi:10.3390/vaccines10060892_

Round 1

Reviewer 1 Report

OVERVIEW:   This article describes adverse events of COVID vaccinations in a population of solid cancer patients undergoing treatment in Pisa Italy. It may or may not be very novel. Please consider the referneces below carefully to determine novelty. Regardless of novelty, it is important to substantiate the safety of COVID-19 vaccination, and this article may offer some validation or additional insight into the results from these previous articles. If the article’s conclusions largely mirror those of these previous publications, it may make sense to mention the specific aspects of the population that make it different or validating in the Abstract, for example, by adding that the population is from Italy (or perhaps “Pisa, Italy”). Vaccine safety is a very important topic and has broad public health implications, An extra article that validates and emphasizes and clarifies vaccine safety is welcome. Please pay attention to my specific comments (in the PDF file submitted) to make sure that this article is conveyed with a tone and style and accuracy that best advances public health interests.  

LITERATURE CHECK:   The manuscript needs to cite these articles, that have performed similar or rleated analysis. Please compare and contrast and contextualize your results in light of these references:  

Desai A, Gainor JF, Hegde A, Schram AM, Curigliano G, Pal S, Liu SV, Halmos B, Groisberg R, Grande E, Dragovich T, Matrana M, Agarwal N, Chawla S, Kato S, Morgan G, Kasi PM, Solomon B, Loong HH, Park H, Choueiri TK, Subbiah IM, Pemmaraju N, Subbiah V; COVID19 and Cancer Clinical Trials Working Group. COVID-19 vaccine guidance for patients with cancer participating in oncology clinical trials. Nat Rev Clin Oncol. 2021 May;18(5):313-319. doi: 10.1038/s41571-021-00487-z. Epub 2021 Mar 15. Erratum in: Nat Rev Clin Oncol. 2021 Mar 23;: PMID: 33723371; PMCID: PMC7957448.  

Funakoshi Y, Yakushijin K, Ohji G, Hojo W, Sakai H, Takai R, Nose T, Ohata S, Nagatani Y, Koyama T, Kitao A, Nishimura M, Imamura Y, Kiyota N, Harada K, Tanaka Y, Mori Y, Minami H. Safety and immunogenicity of the COVID-19 vaccine BNT162b2 in patients undergoing chemotherapy for solid cancer. J Infect Chemother. 2022 Apr;28(4):516-520. doi: 10.1016/j.jiac.2021.12.021. Epub 2021 Dec 30. PMID: 35090826; PMCID: PMC8716153.  

Shulman RM, Weinberg DS, Ross EA, Ruth K, Rall GF, Olszanski AJ, Helstrom J, Hall MJ, Judd J, Chen DYT, Uzzo RG, Dougherty TP, Williams R, Geynisman DM, Fang CY, Fisher RI, Strother M, Huelsmann E, Adige S, Whooley PD, Zarrabi K, Gupta B, Iyer P, McShane M, Yankey H, Lee CT, Burbure N, Laderman LE, Giurintano J, Reiss S, Horwitz EM. Adverse Events Reported by Patients With Cancer After Administration of a 2-Dose mRNA COVID-19 Vaccine. J Natl Compr Canc Netw. 2022 Feb;20(2):160-166. doi: 10.6004/jnccn.2021.7113. PMID: 35130494.  

Thomas SJ, Perez JL, Lockhart SP, Hariharan S, Kitchin N, Bailey R, Liau K, Lagkadinou E, Türeci Ö, Åžahin U, Xu X, Koury K, Dychter SS, Lu C, Gentile TC, Gruber WC. Efficacy and safety of the BNT162b2 mRNA COVID-19 vaccine in participants with a history of cancer: subgroup analysis of a global phase 3 randomized clinical trial. Vaccine. 2022 Mar 1;40(10):1483-1492. doi: 10.1016/j.vaccine.2021.12.046. Epub 2021 Dec 24. PMID: 35131133; PMCID: PMC8702495.  

Trillo Aliaga P, Trapani D, Sandoval JL, Crimini E, Antonarelli G, Vivanet G, Morganti S, Corti C, Tarantino P, Friedlaender A, Belli C, Minchella I, Locatelli M, Esposito A, Criscitiello C, Curigliano G. Safety of COVID-19 mRNA Vaccines in Patients with Cancer Enrolled in Early-Phase Clinical Trials. Cancers (Basel). 2021 Nov 20;13(22):5829. doi: 10.3390/cancers13225829. PMID: 34830983; PMCID: PMC8616209.  

Please see the attached document for other comments that need to be addressed.

Author Response

Dear Reviewer, thank you for providing your valuable feedback and insightful comments. Yu'll find attached the responses to your comments.

Reviewer 2 Report

Thanks for an interesting and relevant manuscript investigating the safety of COVID-19 vaccines in cancer patients.

I have only a few points, which could clarify and strengthen the manuscript:

Line 78: It is unclear, what the 7 days period refers to.

Line 96-98: It is unclear if/how model assumptions, especially of the linear regression (e.g. normality of residuals) were checked.

Line 107: Proabably there was a maximum number of days after the vaccine, for which adverse events were considered related to the vaccine. This maximum time, should be stated.

Line 120-122: It is unclear what the rest of the 33 COVID-19 cases experienced. 2 patients + 36.4% asymptomativ + 39.4% mild does not give a total of 100%/33 pts.

Table 1 (and in general): There is inconsistency between using , and . for decimal points (e.g. 62.3 or 50,7). Moreover, it would be helpfull with % signs and a constant number of decimal places for the statet percentages.

Line 162-175: I would suggest moving this section to the methods section.

Line 177: It is unclear if the OR=0.552 implies that young age is protective or a risk factor (it is more clear from Table 2, but not in the text).

Table 2: The RC column could be removed, as the information is contained in the OR, which is easier to interpret.

Table 3+4: Reporting regression coefficients with 95% CI would be more informative than reporting PCCs.

Line 223-224: It is unclear from this statement if the adverse avents in your cohort are more or less common than in the healthy population.

Author Response

(The authors gave the same response as above.)

Reviewer 3 Report

Thank you for inviting me to review this manuscript. This article contains important findings as patients with serious comorbid conditions were limited in the original clinical trials. I have some comments which should be considered before making any decision on the manuscript.

  1. The rationale of the study is not clear. Please explain in the introduction section why the safety profile of COVID-19 vaccines is needed among active cancer patients?
  2. The methodology section of this manuscript is very limited and general. I will suggest authors consider the following headings in the manuscripts; ethics, study location and design, study population with inclusion and exclusion criteria, study instrument (survey), validation and reliability of the study instrument, components of the study instrument, data collection, outcome variables, data analysis.
  3. Please explain the grading system of side effects in the method section.
  4. I noticed that the authors did not adjust the treatment during the logistic regression analysis. Since the patients are receiving various modalities of cancer treatment, some of the side effects might be associated with the drugs/therapy/or previous traditions. Pain, fatigue, myalgia, and changes in body temperature frequently occur among cancer patients. How the impact of other factors causing these symptoms can be excluded from the analysis. There is a need to clarify the analysis section which must be discussed in the discussion.
  5. The authors did not discuss other comorbid conditions and they were not considered in the analysis too. 
  6. There is no detail on ECOG PS in the method section.
  7.  a substantial part of our patients is on treatment with analgesic drugs, corticosteroids, etc. which could hide any side effects. Can the authors provide the details and proportion of such treatment?
  8. Please confirm that no medication or additional therapy was started among patients due to the side effects of the COVID-19 vaccines.
  9. Authors have explained that the incidence of side effects concord with what was reported in the clinical trials. I have a suggestion to compare the incidence of side effects with other studies conducted in Italy and other parts of the world. 

Author Response

Comments from Reviewer 3

Comment 1:  The rationale of the study is not clear. Please explain in the introduction section why the safety profile of COVID-19 vaccines is needed among active cancer patients?

Response: thank you for pointing this out, we agree with this comment. Therefore, we have modified the final section of the introduction to better understand the rational of our study (lines 78-82)

Comment 2: The methodology section of this manuscript is very limited and general. I will suggest authors consider the following headings in the manuscripts; ethics, study location and design, study population with inclusion and exclusion criteria, study instrument (survey), validation and reliability of the study instrument, components of the study instrument, data collection, outcome variables, data analysis.

Response: We fully agree with this and have modified methodology section according to your suggestion, providing more detailed information regarding study methods and dividing the section in subheadings (lines 83-158).

Comment 3 Please explain the grading system of side effects in the method section.

Response: we agree with this and moved it in the method section (lines 119-133)

Comment 4 noticed that the authors did not adjust the treatment during the logistic regression analysis. Since the patients are receiving various modalities of cancer treatment, some of the side effects might be associated with the drugs/therapy/or previous traditions. Pain, fatigue, myalgia, and changes in body temperature frequently occur among cancer patients. How the impact of other factors causing these symptoms can be excluded from the analysis. There is a need to clarify the analysis section which must be discussed in the discussion.

Response: Thank you for your comment. We dichotomize treatment variable in two categories (immunotherapy-based treatment and non-immunotherapy treatment) and It was properly included in all the regression analysis.

We choose to include just the variable “treatment with immunotherapy” in the regression models because the other anticancer treatments (target therapies and chemotherapies) have similar cytotoxic mechanism of action and similar spectrum of adverse events. Instead, we thought that immunotherapy, due to its intrinsic mechanism of eliciting immunity response, leads a completely different spectrum of side effects than other anticancer treatments.

According to your suggestion, we performed other analysis including the other different categories as anticancer treatment and did not emerge any statistical significance. According to this analysis we updated the discussion (lines 414-415)

Comment 5 The authors did not discuss other comorbid conditions and they were not considered in the analysis too.

Response: morganti – thank you for your comment, we analyzed and categorize comorbidities as shown in table 3,4,5

Comment 6 There is no detail on ECOG PS in the method section.

Response: we add this information in the method section (line 107)

Comment 7 a substantial part of our patients is on treatment with analgesic drugs, corticosteroids, etc. which could hide any side effects. Can the authors provide the details and proportion of such treatment?

Response: Thank you for this suggestion. It would have been interesting to explore this aspect. However, unfortunately we are not able to recover these data from medical records within the required time by Editor for the revision process

Comment 8 Please confirm that no medication or additional therapy was started among patients due to the side effects of the COVID-19 vaccines.

Response: Thank you for pointing this out, percentages of people requiring symptomatic treatment is an important information and according to your suggestion we added it in the manuscript (lines 202-203, 220-221)

Comment 9 Authors have explained that the incidence of side effects concord with what was reported in the clinical trials. I have a suggestion to compare the incidence of side effects with other studies conducted in Italy and other parts of the world.

Response: We agree with your suggestion. Although a direct formal comparison between different studies is not feasible, we tried to emphasize this point contrasting and contextualizing our results in light of other added mentioned papers (Funakoshi et al lines 380-385, Shulman et al lines 386-389, Thomas et al lines 341-344, Trillo Aliaga et al lines 390-393)

Round 2

Reviewer 1 Report

The authors have addressed my concerns.

Reviewer 3 Report

The authors have addressed all my suggestions. I have no more concerns.